# Understanding money-management behaviour and its potential determinants among undergraduate students: A scoping review

**Theepa Cappelli, Adrian P. Banks, Benjamin Gardner** *

School of Psychology, University of Surrey, Guildford, United Kingdom

* benjamin.gardner@surrey.ac.uk

## Abstract

### Background

University students typically face acute financial pressure, which can adversely impact mental health, wellbeing, and academic outcomes. This scoping review of qualitative and quantitative studies aimed to identify distinct money-management behaviours, and psychological determinants, to inform future interventions.

### Methods

Two electronic databases were searched for observational studies focusing on money-management behaviours and their correlates (in quantitative studies) or reflections on experiences of such behaviours (qualitative studies). Of 789 unique papers identified, 12 papers, reporting 10 distinct studies (six quantitative, two qualitative, two mixed-methods), were entered into review. We inductively categorised all behaviours and psychological correlates, and narratively synthesised findings.

### Results

We documented 15 distinct money-management behaviours, which fitted five higher-order categories: budgeting, saving (i.e., building funds), spending, borrowing, and settling debts. Twenty-two distinct potential correlates were observed, which fitted six categories: personality characteristics, financial beliefs and knowledge, attitudes, affective responses, self-efficacy and control, and social influences. Financial beliefs and knowledge, attitudes, self-efficacy and control, and social support from parents and peers were generally associated with 'better' money management practices.

### Conclusion

Heterogeneity in behaviours and correlates studied precluded definitive conclusions. Future studies should more comprehensively adopt theories and concepts from behavioural science, to distinguish between different money-management behaviours, identify which

**Data Availability Statement:** All relevant data are within the manuscript and its Supporting Information files.

**Funding:** The author(s) received no specific funding for this work.

**Competing interests:** The authors have declared that no competing interests exist

behaviours have most impact on students, and establish which specific determinants are most related to which money-management behaviours.

## Introduction

University students worldwide are facing increasing financial pressure. In the UK, higher education has become substantially more expensive in recent decades, with students leaving university with significantly higher debt than at the turn of the century [1–3]. Notable changes over this period have included increasing tuition fees for UK students [4, 5], higher inflation and increased cost of living [6, 7]. Growing concerns around student living costs, tuition fees and loan burdens have also been documented in Australia and Asia [8, 9]. In the US, it has been argued that rising debt levels among graduates, who typically take student loans, may both discourage potential applicants and motivate graduates to seek career paths based on speedier debt repayment [10]. For most students, starting university presents a cluster of major life transitions: typically, full-time students live for the first time away from home, and must adapt to newfound financial independence alongside new social and learning environments [11, 12]. Autonomously managing day-to-day spending requires students to negotiate multiple and potentially competing expenses, including rent and living costs, groceries, transport, tuition fees, social activities and other everyday expenses [13, 14]. University students often lack knowledge and experience in managing their money [15] and tend to make relatively poor financial decisions [16, 17]. Given also that students typically have little disposable income, money-management is often stressful for students, and can adversely impact their mental health and wellbeing [5, 18, 19]. Financial pressure can also lead to poorer academic performance [20]. While upstream structural changes may be needed to tackle the root causes of financial stress among students, behaviour change interventions that encourage students to manage their money optimally–by, for example, budgeting, or minimising unnecessary expenditure–may also be beneficial.

Developing effective money-management interventions for students requires an understanding of students' money-management behaviours and their determinants [21]. Interventions should ideally be designed systematically, whereby knowledge surrounding the psychological determinants of that behaviour is used to select techniques to change behaviour via those determinants [22]. Psychological theory should be used to identify which determinants to focus on, and to organise these into a coherent framework and thereby generate hypotheses around how best to change behaviour [23]. Yet, much of the evidence regarding students' money-management has focused on financial knowledge or literacy as outcomes of interest, rather than on specific financial behaviours [24–26]. Where studies have assessed student money-management behaviour, operationalisations of behaviour appear to have varied. For example, some research on 'money-management' or 'financial behaviour' among students has focused on specific behaviours such as budgeting [27], whereas other studies have focused on a broader range of actions [e.g., 14, 28]. Discrepancies among conceptualisations of 'money-management behaviour' makes it difficult to draw intervention design conclusions. Synthesising evidence surrounding the determinants of behaviour is an important preliminary step in intervention design [e.g., 21, 23]. Where evidence in a domain is heterogeneous, evidence synthesis should focus on summarising how a concept has been operationalised [29].

Scoping reviews aid intervention development by identifying and organising evidence surrounding how a target behaviour has been studied to date [29]. Whereas a 'traditional'

systematic review would typically aim to establish the strength of relationships between a behaviour and its determinants, a scoping review aims to document what evidence exists in the area [29, 30]. By providing an assessment of the breadth and depth of a particular research field, scoping reviews can identify knowledge gaps that must be filled prior to intervention development [31].

The present study reports a scoping review designed to establish how money-management behaviour and its psychological determinants have been assessed in observational studies of university students. We reviewed both quantitative evidence, which has the potential to reveal statistical relationships between behaviour and its determinants, and qualitative evidence, which can reveal important experiences and meanings surrounding money-management among students. While intervention development requires understanding the psychological *antecedents* of behaviour, we reviewed evidence relating to any potential psychological *correlate* of behaviour to allow for evidence to be extracted from cross-sectional or retrospective studies.

The review aimed to address three research questions, one of which was operationalised differently for quantitative and qualitative studies:

1. Which money-management behaviours have been studied among university students?

2a. Which psychological correlates of students' money-management behaviour have been assessed in quantitative studies?

2b. Which experiences surrounding money-management behaviour have been explored in qualitative studies?

3. Which psychological theories have been used to understand students' money-management behaviours, and correlates and experiences surrounding those behaviours?

## Materials and methods

This review is structured using applicable components of the Joanna Briggs Institute's (JBI) framework [31], and was conducted using appropriate items from the PRISMA Extension for Scoping Reviews [32]. We did not publish a protocol for this study prior to execution of the review.

### Identifying relevant studies

**Study eligibility criteria.** Studies were eligible for inclusion where they were: (a) published as full-text (b) in peer-reviewed journals (c) from 2000 onwards, (d) in either English or German, and (e) reported primary empirical (quantitative, qualitative or mixed-methods) data (f) among undergraduate students (g) focusing on money-management behaviour, and either (h) psychological correlates of typical or actual money-management behaviour (in quantitative studies) and reported statistical relationships with typical or actual behaviour or (i) experiences surrounding typical or actual money-management behaviour (in qualitative studies).

Our search allowed any studies of 'young adults' to be retrieved, but we excluded any studies from which an undergraduate-only sample could not be isolated. German studies were eligible because the first author (TC) is fluent in both German and English. Studies that assessed outcomes of behaviour (e.g., credit card debt), but not behaviour itself (frequency of credit card use), were excluded [e.g., 33].

**Search procedure.** Studies were identified using two online databases (PsychInfo, ProQuest Advance). Search terms were created to specify the target *population* (i.e "student" OR

"young adult" OR "undergraduate" OR "university" OR "college" OR "freshman") and *behaviour* of interest ("day-to-day spending", "spending decision", "money related behaviour", "money management", "spend", "money", "finance"). To avoid unnecessarily restricting the search, terms were not specified for outcomes, correlates, or study type. The search string combined the terms above with the AND operator, and was used to search title, abstract, keywords and subject fields. The latest search was undertaken in May 2024.

## Study selection

Screening was undertaken by author TC. Of 1,005 papers identified in the database search, 226 were duplicates, and 748 were removed based on title and abstract screening. Of the remaining 41 papers, 29 were removed after full-text review, leaving 12 papers. In two instances, two papers each reported analyses of the same dataset, so were treated as single studies for review purposes [14, 25, 33, 34]. Thus, the 12 retained papers reported a total of 10 studies (see Fig 1).

## Extracting the data

The following data were extracted from all studies: author and year of publication; setting (country); sample (description and size); study type; behaviours relating to which data were collected; and whether and which psychological theories were used and how. For quantitative studies, and quantitative components of mixed-methods studies, we extracted data on the psychological correlates studied, the items used to measure those correlates, and statistical relationships between each correlate and behaviour. Where a correlate was measured but data were unavailable regarding whether or to what extent this variable was associated with behaviour, no data relating to this variable were extracted. For qualitative studies and qualitative components of mixed-methods studies, we meta-synthesised authors' commentaries on key psychological concepts surrounding money-management behaviours. We use direct quotations from the data to illustrate these.

## Collating, summarising and analysing the results

Data from the 10 studies were collated using a descriptive-analytical approach [34]. For all studies, data relating to the behaviour(s) of interest were tabulated, and subsequently categorised inductively to identify and define discrete types of money-management behaviour. Similarly, potential correlates assessed in quantitative studies and phenomena from qualitative studies were categorised into clusters, which we inductively defined. Relationships between psychological correlates and behaviours derived from the quantitative data are reported via a narrative synthesis. In this narrative synthesis, we describe behaviours according to the labels we assigned to them, rather than using authors' own descriptions. To summarise statistical relationships between potential correlates and behaviour, we report correlation coefficients (Pearson's r) where these were provided, as these coefficients establish the *size* of the relationship between two variables. Correlation effect sizes were established using Cohen's [35] criteria, where $r \approx .20$, .50 and .80 represent small, medium (moderate) and large (strong) effects respectively. Where correlation coefficients were not provided, we report relationships based on statistical significance (at $p < .05$), though this reveals only *whether* a relationship was observed, not the size of that relationship.

Data on key psychological concepts from the qualitative studies were categorised using qualitative meta-synthesis methods and are reported via a standalone narrative synthesis. All other data were descriptively analysed.

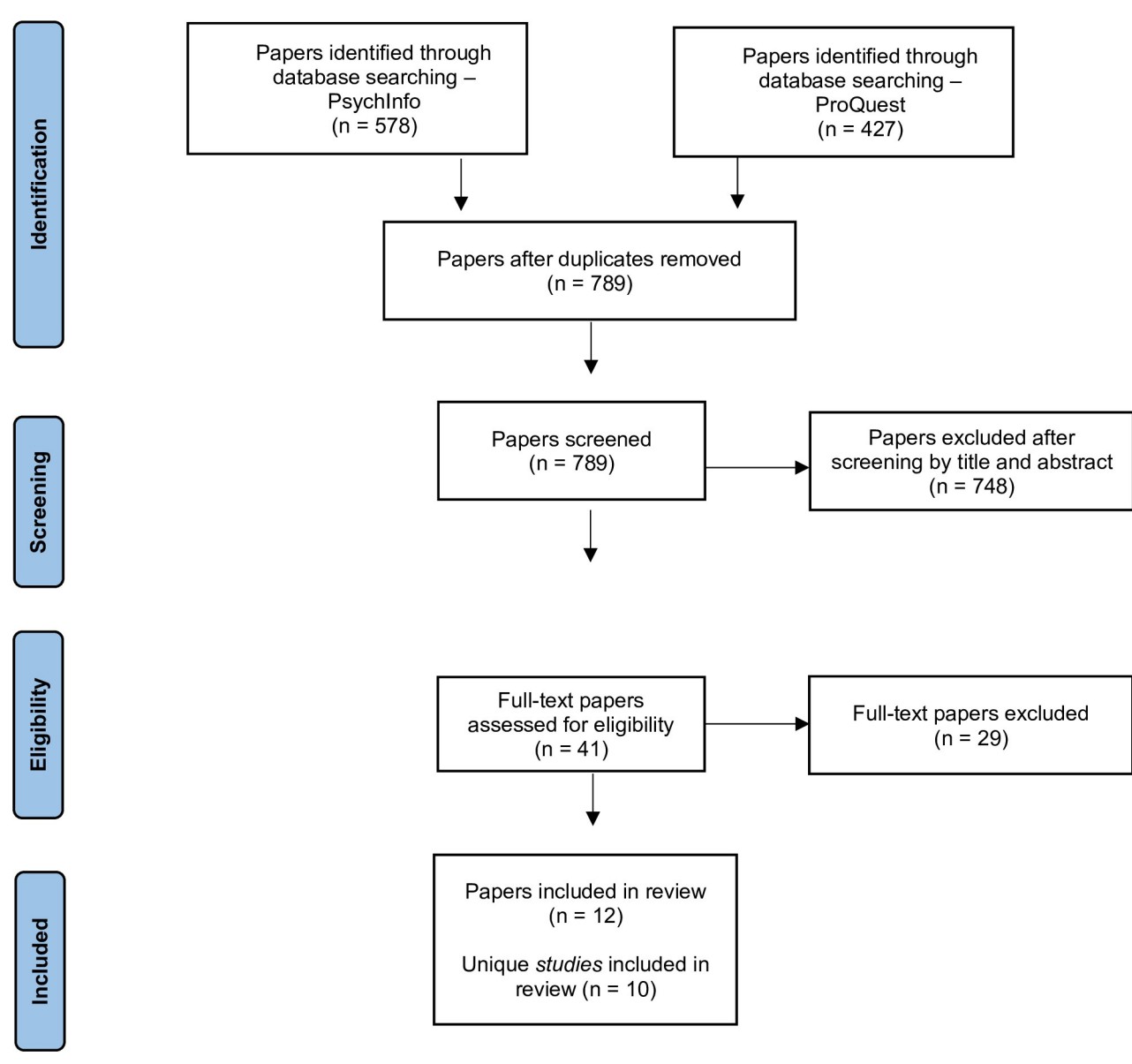

**Fig 1. PRISMA flow chart.**

## Results

### Study characteristics

**Setting and sample characteristics.** Of the ten studies, four were undertaken in the USA, two in Indonesia, and one in each of Australia, Hong Kong, Malaysia, and Sri Lanka. Participant age ranged from 18–40 years. Nine studies focused on undergraduate students only, and one focused on both undergraduates and postgraduates, but only undergraduate data are reported here [36].

**Study designs.** Six studies used quantitative data only and were cross-sectional in nature; two studies used qualitative data only, with one using focus group data only and the other combining focus group and semi-structured interview data; and two studies used cross-sectional mixed-methods sequential designs, whereby quantitative data were collected (via

questionnaires and surveys) prior to qualitative data collection. In one mixed-methods study, qualitative data were collected via open-ended, free-text responses to questionnaires, and in the other study, focus group. The 10 studies thus comprised a corpus of eight quantitative datasets and four qualitative datasets for review. All data in all 10 studies were self-reported.

Mean sample size was 392 in the quantitative-only studies, and 40 in the qualitative-only studies. In the mixed-methods studies, mean sample size was 474 in the quantitative component, and sample size for the qualitative component was only reported in one study [N = 28; 37]. All six quantitative-only studies aimed to understand the correlates of students' behaviours.

**Which money-management behaviours have been studied among university students?.** Across the 10 studies, we identified 15 discrete subtypes of 'money-management behaviour', which we organised into five categories (Table 1). The 'budgeting' category included five behaviours that described some element of planning or reflecting on expenditure. At least one 'budgeting' behaviour was assessed in three quantitative-only, one qualitative-only and one mixed-methods study. 'Borrowing' encompassed three distinct behaviours, which were assessed across two quantitative-only and one mixed-methods studies. 'Settling debts' incorporated three behaviours, which were assessed across four quantitative-only and one qualitative-only study. 'Spending' included two behaviours, which were assessed in two quantitative-only, one qualitative-only and one mixed-methods study. 'Saving', which we defined in relation to the accumulation of funds (rather than minimising expenditure), incorporated two behaviours, assessed in two quantitative-only and one qualitative-only study. Six of the eight quantitative datasets used behaviour measures that integrated multiple behaviours (e.g., budgeting and settling debts).

## Which psychological correlates of students' money-management behaviour have been assessed in quantitative studies?

Across the eight quantitative datasets, we identified 22 distinctive psychological correlates, which we clustered into six categories: Personality Characteristics, Financial Beliefs and Knowledge, Attitudes, Affective Responses, Self-Efficacy and Control, and Social Influences (Table 2).

*Personality characteristics* were examined in one of the eight quantitative datasets (13%), which found a small correlation between impulsivity and 'planning finance' (r = -.17).

*Financial beliefs and knowledge* variables were assessed in three of the eight studies (38%). Two out of the three showed that financial literacy had a positive relationship with 'using credit cards' and 'making credit card payments in full'. One study found that 'healthy financial behaviour', as indexed by a combined measure of budgeting, settling debts, and saving behaviours (i.e., tracking expenses, maintaining a budget, making credit card payments in full, saving money, and investing) had a medium sized positive relationship with financial knowledge (r = .48) and a small-to-medium positive correlation with parental direct teaching (r = .43). Another study found that anticipated future income had no relationship with 'using credit cards'.

*Attitude* variables were explored in three of the eight studies (38%). Two studies found a positive attitude-behaviour relationship, such that people with more financially prudent attitudes engaged more in 'healthy financial behaviour' (i.e., combined measure of budgeting, settling debts, and saving behaviours). This relationship was moderate in one study (r = .56), and no coefficient was reported in the other study (p < .05). In another study, attitude towards budgeting had a medium-to-large positive correlation with 'creating or maintaining a budget' (r = .66).

**Table 1. Behaviours and categories of behaviours identified in all reviewed studies, and correlates modelled in quantitative studies.**

| Behaviour | Definition | Studies in which behaviour assessed | Example items from studies | Categories of psychological variables investigated (in quantitative studies) as correlates of this behaviour |
|---|---|---|---|---|
| **Budgeting** | | | | |
| Creating or maintaining a budget | Encompasses various activities oriented around planning or reflecting on expenditure, including expense tracking, ensuring that expenditure does not exceed income (i.e., 'living within means'), and managing income and expenditure. | [11, 24, 38–41] | *"I have a weekly (or monthly) budget that I follow (always–never)"* [40] *"In the past three months, how often have you budgeted your finances?"* [24] | Attitude Affective Responses Social Influence Self-efficacy Financial beliefs and knowledge |
| Planning purchases | Planning what to purchase in order to stay within budgetary limits | [39] | *"How often do you plan your spending?"* [39] | Affective responses |
| Tracking expenses | Keeping a record of spending on specific items or total spending | [11, 38–40] | *"Do you keep track of income and expenditure?"* [40] | Affective Responses Self-efficacy Social influence |
| Comparing prices | Evaluating and contrasting the costs of similar products or services, or the same product or service provided by multiple sources | [39, 40] | *"How often do you compare prices before purchasing?"* [39] | Affective Responses Self-efficacy Financial beliefs and knowledge |
| Planning finance | Conscious and deliberate process of managing one's financial resources to achieve personal life goals. | [39] | *"How often do you plan your finances in mind but not writing down a budget (e.g. putting money aside for savings, daily expenses, investment, entertainment, etc.)"* [39] | Personality |
| **Borrowing** | | | | |
| Using credit cards | Paying for goods or services with a credit card. | [11, 38, 40] | *"I prefer to buy things on credit card rather than wait and save up"* [11] | Financial beliefs and knowledge Attitude |
| Use of cash advance on credit card | Borrowing cash from a credit card account. | [11, 38, 41] | *"I obtain cash advances to pay money towards other credit card balances"* [42] | Affective response Personality Attitude |
| Maximising credit card | Borrowing as much money as is permitted on a credit card | [11, 38, 42] | *"I reach the maximum limit on my credit card"* [42] | Affective Responses Attitude |
| **Settling debts** | | | | |
| Settling debts on time | Paying bills, loan instalments or other debts before or when they are due to be paid | [11, 38, 42] | *"Are you able to settle debts on time?"* [42] | Affective responses |
| Making minimal payments on loans | Paying minimal instalments on formal loans or other debts | [39] | *"How often do you make minimum payments on loans?"* [39] | Attitude |
| Making credit card payments in full | Paying credit card bills in full (and thereby avoiding incurring additional charges or penalties, e.g. interest) | [11, 35, 38, 41] | *"I pay off the full credit card outstanding amount every month"* [11] | Social influence Financial beliefs and knowledge |
| **Spending** | | | | |
| Saving (minimising usual expenditure) | Avoiding usual expenditure or buying cheaper alternatives, to conserve money | [41] | *"I save money by skipping dinner"* [41] | Affective Responses |
| Overspending | Spending more money than is owned or earned, or more money than was budgeted for. | [11, 36, 42] | *"I spend more money than I earn"* [42] *"To keep up with friends, I spend money even when I cannot afford it"* [36] | Affective Responses Social Influences |
| **Saving (building funds)** | | | | |
| Accruing money | Encompasses various activities involved in accruing money over time, such as depositing into a bank account or investing | [11, 35, 38, 41] | *"I set financial goals and objectives in my life"* [11] | Social Influences Financial beliefs and knowledge |
| Investing | Spending money on a product expected to grow in value, so as to build wealth | [11, 38] | *"Do you engage in investing for long-term financial goals regularly?"* [11] | Social influences |

**Table 2. Psychological correlates observed in eight quantitative datasets.**

| Determinants | Definition of the concept as assessed in the reviewed studies | Number of datasets in which assessed |
|---|---|---|
| *Personality Characteristics* | | |
| Impulsivity | A tendency to make spontaneous financial decisions without considering potential long-term consequences. | 1 dataset (13%) |
| *Financial Beliefs and Knowledge* | | |
| Financial literacy | Financial literacy is knowledge and understanding of financial concepts and risks, as well as the skills and attitudes to apply such knowledge and understanding in order to make effective decisions across a range of financial contexts, to improve the financial well being of individuals and society, and to enable participation in economic life. | 2 datasets (25%) |
| Financial knowledge | An overall understanding of financial management concepts. | 1 dataset (13%) |
| Parental direct teaching | Perception of the extent to which parents directly taught the student about financial management concepts or behaviours. | 1 dataset (13%) |
| *Attitudes* | | |
| Anticipated income | Forecasted earnings in the future | 1 dataset (13%) |
| Attitude towards debt | A positive or negative evaluation of the concept of borrowing and managing owed money. | 1 dataset (13%) |
| Positive financial attitude | An individual's mindset or disposition towards money and financial decision-making, reflecting their beliefs, values, and emotional responses related to financial matters (how personal biases and perceptions influence financial behaviour) | 2 datasets (25%) |
| Positive attitude towards budgeting | An individual's cognitive, affective, and behaviour towards budgeting. This encompasses their beliefs about the significance and utility of budgeting in managing finances, their emotional response to the process of budgeting, and the behaviours they exhibit in relation to adhering to a budget. | 1 dataset (13%) |
| *Affective Responses* | | |
| Emotional evaluation of favourable social comparison | Positive affective responses to favourable comparisons between one's own and unspecified other's general financial behaviour. | 1 dataset (13%) |
| Anxiety | Stress or worry about one's current or anticipated future financial situation | 2 datasets (25%) |
| Financial wellbeing | Wellbeing arising from or relating to financial status. | 1 dataset (13%) |
| Budgeting affect | The feelings and attitudes people have towards the process of budgeting, including their motivation, stress levels, and overall satisfaction with financial management–how individual's perceive and engage with budgeting which influences their overall financial behaviour and decisions. | 1 dataset (13%) |
| *Self-Efficacy and Control* | | |
| Perceived control | Individual's belief in their ability to manage and influence their financial situation effectively. | 1 dataset (13%) |
| Perceived financial ability | An individual's self-assessment of their financial knowledge and skills, which can include their confidence in managing personal finances, making financial decisions, and understanding financial concepts. | 1 dataset (13%) |
| *Social Influences* | | |
| Social media exposure | Exposure to financial information, trends, opinions or advice encountered on social media platforms. | 1 dataset (13%) |
| Peer influence | The potential influence of peers on students' own financial behaviours, attitudes and decisions. | 1 dataset (13%) |
| People and media norm | Others or media financial opinions, decisions and actions which affects an individual. | 1 dataset (13%) |
| Positive parental descriptive norm / Positive parental norm | Perceptions of positive financial behaviour among parents, which includes tracking monthly expense, spending withing budget, paying credit card balances in full each month, saving money each month for the future and investing for long-term financial goals. | 2 datasets (25%) |
| Parental subjective norm | Perceived approval from parents for positive financial behaviour | 1 dataset (13%) |
| Parental relationship | The parental financial relationship refers to the interaction and dynamics between parents and their college-aged children regarding financial matters, influencing the child's well-being and adjustment to adult life. | 1 dataset (13%) |
| Communication with parents | Communication with parents regarding financial behaviour or practices. | 1 dataset (13%) |

*Affective responses* were assessed in three studies (38%). Two studies exploring anxiety respectively showed that higher anxiety was associated with higher 'overspending' (p < .02) and 'maximising credit cards' (p < .04; coefficients not reported). A combined measure of budgeting, settling debts, and saving behaviours showed a null relationship with anxiety (r = -.09), and a small positive correlation with financial well-being (r = .14). One study found that negative emotional responses to managing personal finances had a small negative relationship with 'maintaining a budget' (r = -.25).

*Self-Efficacy and Control* variables were assessed in three of the eight studies (38%). Two studies focused on perceived financial ability. One found perceived ability to have a small positive correlation with 'budgeting' (r = .26), and another showed a small-to-medium positive relationship with 'healthy financial behaviour' (i.e. combined measure of regularly maintaining budget, saving, tracking expenses, use of credit card; r = .44). One study showed financial efficacy and financial controllability had a negative relationship with a combined measure of 'using credit cards' and 'maximising credit cards' (p's < .01).

*Social Influences* were assessed in four of the eight studies (50%). One study showed that exposure to finance social media users and supportive peers was positively associated with a combined measure of budgeting, settling debts, and saving behaviours (p < .05). Another study found that parental norms were positively related with a combined measure of budgeting, settling debts, and saving behaviours (p < .001). One study of parental influences found that a combined measure of budgeting, settling debts, and saving behaviours showed a small positive correlation with parental subjective norms (r = .18) and small-to-medium positive correlations with parental descriptive norms (r = .34) and parental relationships (r = .26). Another study found that communication with parents and peers about finances had a positive relationship with 'creating and maintaining a budget' and 'accruing money', though no coefficients were reported.

## Which experiences surrounding students' money-management behaviour have been observed in qualitative studies?

**Financial beliefs and knowledge.** Two of four qualitative datasets (50%) explored how students acquired money-management knowledge and skills. These focused on learning from parents, via direct teaching about how and why to manage their money efficiently (e.g., *"I was disciplined by my parents as a little kid not to overspend"*; 23, p301). Some participants discussed vicarious learning from observing parents' poor money-management:

> "*My father is an over-spender [. . .]. Because of him, I have learnt not to spend more than I earn*" [42, p176]

Other described learning money-management skills based on personal experience of suboptimal money-management ("*the best teacher is your own mistakes*"; [42, p180]).

**Social influences.** Three studies (75%) focused on the perceived importance of spending, often beyond means, for socialising with peers. Students felt that spending money on social activities was essential for maintaining relationships, and that choosing not to spend money on such activities risked social exclusion. Some students described striving to meet spending norms as a means of affiliating with others (*"I'll kind of match whatever they [friends] are spending or willing to [spend]"*; [37, p205]). Some students, however, sought to socialise with friends with similar financial backgrounds and outlooks ("*I don't associate with people . . . with higher incomes*"; [25, p213]).

Social media was also mentioned by some students as an influence on their money-management, with some students spending to match latest trends, or mimic the lifestyle or look of influencers.

Religion was cited as an influence on money-management in one study, with some students feeling compelled to live frugally in line with religious beliefs.

**Affective responses.**   Negative affect, and in particular stress, was explored in two of the four studies (50%). One study observed that students who displayed careful, controlled money-management behaviour reported experiencing distress in response to expenses outside of their control [40]. Students reported a bidirectional and potentially spiralling relationship between negative affect and spending, with experiences of stress, anxiety and guilt prompting spending, which in turn could worsen negative affect (*"I was freaking out last night because of a bad day, I spent everything on eBay. [. . .] I manage guilt by buying more stuff"*; [43, p302]).

**Self-efficacy and control.**   One study (25%) focused on perceptions of control over spending across different spending methods. Some students felt confident in managing and controlling their money by using credit cards, as this could itemise *where* money was spent, whereas others preferred cash as they felt it gave them more control over *how much* they spent ("*I prefer cash because you know how much you are spending whereas with a card, it feels unlimited"*; [43, p300]). Some students felt that attempting to manage their money was futile, because external factors were the main influence on their spending ("*we really don't have that much control over, you know, um where our money goes"*; [43, p299]).

## Which psychological theories have been used to understand students' money-management behaviours, correlates and experiences?

Only four studies explicitly referred to theory, all of which were quantitative studies that used theory to identify variables to assess as potential correlates of money-management behaviours. The theories used were the Theory of Planned Behaviour [TPB; 44] and its predecessor, the Theory of Reasoned Action [45], Consumer Socialisation Theory [40], and Money Management Model [a theory developed in previous money-management research; 24]. Only one study sought to comprehensively apply a theory [i.e., the TPB; 38]; the remaining three studies tested only select parts of a focal theory.

## Discussion

Financial pressure among students has the potential to adversely impact mental health and wellbeing and academic performance. The development of effective interventions to support students to manage their money effectively will be aided by understanding specific money-management behaviours and their psychological determinants. This scoping review was undertaken to identify how money-management behaviour has been studied to date, and document psychological correlates and students' experiences surrounding money-management. We identified ten observational studies of student money-management behaviour, across which 15 discrete behaviours, within five distinct categories, were identified. We also observed six categories of psychological variables surrounding money-management. Variables pertaining to financial beliefs and knowledge, attitudes, self-efficacy and control, and social support from parents and peers were generally associated with 'better' money management practices. However, heterogeneity in the behaviours and correlates studied precludes definitive conclusions regarding which specific variables are consistently predictive of which specific behaviours. We documented underuse of psychological theory to organise and understand the determinants of money-management behaviours. Our findings suggest that terms such as 'money-management behaviour' or 'financial behaviour' are too broad. Future studies should seek to identify which specific forms of 'money-management behaviour' have most impact on mental health, wellbeing, academic and other outcomes, and which psychological determinants are most closely related with which money-related behaviours. Existing theoretical

frameworks should be drawn on to identify a more comprehensive range of psychological determinants of money-management behaviours, and test hypotheses regarding how these may interact to determine students' specific money-management behaviours.

The term 'money-management behaviour' is often implicitly used to denote a single, unidimensional behaviour [25, 35, 39, 40, 46]. However, we identified 15 distinctive 'money-management behaviours' that have been studied in observational studies to date, and these could be sorted into five categories: budgeting, saving (building funds), spending borrowing, and settling debts. 'Money-management' thus encompasses a comprehensive range of practices [33], but few studies have explicitly conceived of these as distinct behaviours with potentially distinct determinants. Indeed, the behaviour measures used in most of the reviewed studies combined multiple actions, such as budgeting and building funds. Using such measures as outcomes precludes conclusions regarding which variables predict which specific behaviours. It is important that distinct behaviours are conceived of and studied distinctively, for several reasons. First, different money-management behaviours may differ in their impact on students' health and wellbeing, academic and other outcomes [47]. For example, the ability to accumulate funds is closely linked to long-term financial security and has a protective effect against future stress, while settling debts on time can manage more immediate stressors and so protect wellbeing in the shorter term [48–50]. Conversely, excessive borrowing and mismanagement of credit cards has been found to prompt anxiety and worsen academic performance [39, 51]. Second, and relatedly, the impact of behavioural money-management interventions for students on outcomes such as health and wellbeing will vary according to which behaviour they target. Further research is needed regarding which specific money-related behaviours have greatest impact on students' health and wellbeing, academic performance, and other outcomes.

We identified 22 discrete psychological variables examined as correlates of a money-management behaviour across eight quantitative datasets, and additional potential determinants and sequelae of money-management from four qualitative datasets. We clustered these into six categories, spanning personality characteristics, financial beliefs and knowledge, attitudes, affective responses, self-efficacy and control, and social influences. We found statistical relationships between variables within each of these categories and at least one form of money-management behaviour within the reviewed quantitative data. For example, impulsivity was associated with overspending [39], which resonates with broader psychological literature suggesting that personality is a core predictor of decision-making [52]. Similarly, affective responses, including anxiety and positive emotions, have been linked to overspending and budgeting [24, 42]. This aligns with psychological theory showing that emotional states can significantly impact decision-making processes [53]. All quantitative studies included in the review were cross-sectional in nature, which precludes conclusions around the causal direction of relationships between variables. However, insights from the qualitative data suggested that the impact of affective states on some money-management behaviours is bidirectional; specifically, negative affect prompted spending behaviour, which in turn worsened affective states. Together, these findings point both to the limitations of modelling statistical relationships based solely on cross-sectional, observational data, and the advantages of used qualitative evidence to explore and explain statistical findings. Given the dearth of evidence surrounding the determinants of students' money-management behaviours, we recommend that mixed-methods designs are used to optimise insights generated by future studies in this area.

Syntheses of quantitative data showed that variables relating to financial beliefs and knowledge, attitudes, self-efficacy and control, and parental and peer social influence were predominantly related with 'healthier' money management practices, so should be harnessed as mechanisms for interventions to support students to better manage their money. Relationships between these variables and money-management practices appeared inconsistent, however.

This may in part be because quantitative methodologies can oversimplify potentially complex phenomena. For example, parental norms and communication with parents and peers were each found to be positively associated with 'better' money management among students in quantitative studies, yet qualitative studies provided examples of students deliberately adopting positive money-management practices in response to their perceptions that their parents had managed their money badly [42]. Similarly, peer influence can have an adverse influence on money management when students engage in spending behaviours aimed at matching their peers, to maintain social standing [37]. The discrepancy in findings across studies underscores the complexity of social influences on money management behaviour [37, 54], and suggests that further research is needed to explore what determines whether others will have a positive or negative impact on money management. Other observed relationships may have lacked consistency across studies owing to heterogeneity in the specific behaviours and correlates studied. Such inconsistency is perhaps unsurprising given the considerable heterogeneity we observed in methods used to study money-management behaviour and its correlates. We showed that a diverse range of behaviours have been studied, a variety of factors have been modelled as correlates or studied as experiences, and multiple different measures have been used to document these factors. Robust estimation of relationships between two variables requires a consistent operationalisation of those variables, and standardised methods for capturing those variables, such as validated questionnaires.

We observed a lack of coherence in the selection of potential psychological correlates of students' money-management behaviours in the literature to date. Notably, fewer than half of the reviewed studies drew on psychological theory. Using theory allows researchers to build on a comprehensive account of potential determinants and interrelationships between these determinants, and to identify potential psychological pathways through which behaviour may be changed. Of the four studies that used theory, only one tested the theory in its entirety, by modelling the impact of all theorised psychological determinants of risky spending and borrowing behaviours [38]. The remaining three studies sought to model some but not all hypothesised predictors of behaviour. Our findings thus suggest that theory has been under-used in attempts to understand students' money-management behaviours. We recommend that psychological theory is used more consistently and comprehensively in this domain, to better capitalise on existing knowledge regarding the determinants of behaviour.

While theory should ideally be used to inform the selection of variables to study as potential determinants of behaviour, it can also be used retrospectively, to organise determinants that have been studied into a coherent framework [23]. For example, the TPB proposes that behaviour is most closely influenced by intentions to perform the behaviour, which summates all sources of conscious motivation [44]. Intention is determined by three variables–i.e., attitudes towards the behaviour, social norms and perceptions of control over the behaviour–which in turn are shaped by personality characteristics [44]. Variables relating to personality, attitudes and beliefs, social environments, and control perceptions were observed across all studies but, except one notable study that tested the TPB in full [38], these areas have been under-examined in this domain. For example, the qualitative studies we reviewed offered rich coverage of social influences, highlighting the impact of parents, peers, social media, religion and culture, yet quantitative studies have used only a narrow conceptualisation of social pressures. The lack of research attention to the full breadth of motivational influences means that the relative strength of these determinants on specific money-management behaviours remains unclear. The apparent strategy of simply cataloguing potential determinants of behaviour fails to capture complex interplay among these variables; for example, control perceptions can influence attitudinal beliefs, and vice versa [44]. Additionally, only one of the reviewed studies included a measure of intentions, such that it was unclear in the nine other studies how strongly

motivated participants were to engage in specific money-management behaviours. Future observational studies should use theory to ensure a comprehensive range of psychological influences on behaviour are captured.

None of the reviewed studies included any assessment of non-reflective processes. While the TPB focuses on conscious and reflective determinants of behaviour, such as evaluations of the overall favourability of action, dual-process models suggest that behaviour is shaped by a combination of reflective and non-reflective (i.e., automatic) processes. Much human behaviour is driven by automatic processes that can either support or undermine intentional actions [55]. For example, spending habits–i.e., actions cued automatically due to learned cue-behaviour associations [56]–can prompt unplanned purchases, especially among those with low momentary self-control [57, 58]. People may also develop habits that support well-being-conducive money-management behaviours, such as saving [57]. Automatic processes can also overpower conscious motivation, such that some students may momentarily engage in money-management behaviours that conflict with their attitudes, beliefs or values [59]. Future research should focus on the role of non-reflective factors in determining everyday money-related behaviours among students.

Limitations must be acknowledged. We reviewed only observational studies of correlates of, or experiences surrounding, money-management behaviour among undergraduate university students. This excluded research that may offer indirect insights into students' money-management, such as intervention studies [e.g., 60], studies undertaken to predict proximal determinants of behaviour such as intentions [e.g., 61, 62], and studies undertaken among both student and non-student samples from which data relating to undergraduate students could not be isolated [e.g., 57, 63]. However, the development of effective behaviour change interventions requires an understanding of a behaviour change problem as it is experienced by members of the target population and other stakeholders [64].

Another limitation of our research is that we mostly summarised quantitative and qualitative data separately, rather than incorporating these into a mixed-methods synthesis. However, this limitation arises mostly from the lack of consistency in the literature to date. Mixed-methods analyses require a degree of homogeneity within the qualitative and quantitative datasets regarding the core behaviour of interest that was not available in our dataset [65]. Indeed, the limitations of our study call attention to important avenues for further research to better understand students' money-management behaviour, including more consistency regarding behaviours of interest.

## Conclusion

Developing effective interventions to support students' money-management behaviours requires understanding which behaviour or behaviours to target, and which psychological variables determine these behaviours. Financial knowledge, attitudes, control perceptions, and social support variables were observed as correlates of money-management behaviours. Yet, considerable heterogeneity in the behaviours and potential determinants studied in the literature to date precludes clear recommendations for designing interventions to modify specific money-management actions among students. We recommend that researchers identify which specific money-management behaviours to target and why, and use comprehensive psychological theories, such as dual process models, to generate and test hypotheses regarding the determinants of money-management in students.

## Supporting information

**S1 Table. Characteristics of reviewed studies.**
(DOCX)

## Author Contributions

**Conceptualization:** Theepa Cappelli, Adrian P. Banks, Benjamin Gardner.

**Formal analysis:** Theepa Cappelli, Benjamin Gardner.

**Funding acquisition:** Benjamin Gardner.

**Investigation:** Theepa Cappelli.

**Methodology:** Theepa Cappelli, Adrian P. Banks, Benjamin Gardner.

**Project administration:** Theepa Cappelli.

**Supervision:** Adrian P. Banks, Benjamin Gardner.

**Validation:** Benjamin Gardner.

**Writing – original draft:** Theepa Cappelli.

**Writing – review & editing:** Theepa Cappelli, Adrian P. Banks, Benjamin Gardner.

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
