## [Decision Letter · Decision Letter 0]

9 Jun 2024

PONE-D-24-18530Understanding money-management behaviour and its potential determinants among undergraduate students: A scoping reviewPLOS ONE

Dear Dr. Gardner,

Thank you for submitting your manuscript to PLOS ONE. After careful consideration, we feel that it has merit but does not fully meet PLOS ONE’s publication criteria as it currently stands. Therefore, we invite you to submit a revised version of the manuscript that addresses the points raised during the review process.

We look forward to receiving your revised manuscript.

Kind regards,

Botond Géza Kálmán, PhD

Academic Editor

PLOS ONE

Journal Requirements:

3. We note that your Data Availability Statement is currently as follows: All relevant data are within the manuscript and its Supporting Information files

Reviewers' comments:

Reviewer's Responses to Questions

**Comments to the Author**

1. Is the manuscript technically sound, and do the data support the conclusions?

Reviewer #1: Yes

Reviewer #2: Yes

2. Has the statistical analysis been performed appropriately and rigorously? 

Reviewer #1: Yes

Reviewer #2: Yes

3. Have the authors made all data underlying the findings in their manuscript fully available?

Reviewer #1: Yes

Reviewer #2: Yes

4. Is the manuscript presented in an intelligible fashion and written in standard English?

Reviewer #1: Yes

Reviewer #2: Yes

5. Review Comments to the Author

Reviewer #1: The article adequately reviews and contextualizes the examined problem. The adapted method and the interpretation of the results are appropriate. The content is concisely described and contextualized with previous and current theoretical background and empirical research on the topic. I propose a minor revision in the literature review. The authors did not refer to sources that were pioneers in the research of the topic, e.g.

Bakken, M.R. (1966). Money Management Understanding of Tenth Grade Students. University of Alberta. https://archive.org/details/Bakken1966/page/n13/mode/2up

Chen, H., & Volpe, R.P. (1998). An Analysis of Personal Financial Literacy Among College Students. Financial Services Review, 7(2), 107–128. https://doi.org/10.1016/s1057-0810(99)80006-7

Chen, H., & Volpe, R.P. (2002). Gender differences in personal financial literacy among college students. Financial Services Review, 11(3), 289–308. https://go.gale.com/ps/i.do?p=AONE&sw=w&issn=10570810&v=2.1&it=r&id=GALE%7CA149166047&sid=googleScholar&linkaccess=abs

Danes, S.M., & Hira, T.K. (1987). Money Management Knowledge of College Students. Journal of Student Financial Aid, 17(1), 4–16.

I also recommend a more detailed discussion of the problem that the parental model and the opinion of peers have a positive effect on the financial behavior of university students, because according to European research experience, the behavior and opinion of parents and peers have a negative effect on the appropriate financial behavior. This is confirmed by e.g. also lines 285-287.

After these changes, I recommend the article for acceptance.

Reviewer #2: The research deals with a relevant topic, its examination contributes significantly to the understanding of university students' money management behaviour and its possible determinants.

The article adequately reviews and contextualizes the examined problem. The adapted method and the interpretation of the results are appropriate. The content is concisely described and contextualized with previous and current theoretical background and empirical research on the topic.

I propose a minor revision in the literature review. The authors did not refer to sources that were pioneers in the research of the topic, e.g.

Bakken, M.R. (1966). Money Management Understanding of Tenth Grade Students. University of Alberta. https://archive.org/details/Bakken1966/page/n13/mode/2up

Chen, H., & Volpe, R.P. (1998). An Analysis of Personal Financial Literacy Among College Students. Financial Services Review, 7(2), 107–128. https://doi.org/10.1016/s1057-0810(99)80006-7

Chen, H., & Volpe, R.P. (2002). Gender differences in personal financial literacy among college students. Financial Services Review, 11(3), 289–308. https://go.gale.com/ps/i.do? p=AONE&sw=w&issn=10570810&v=2.1&it=r&id=GALE%7CA149166047&sid=googleScholar&linkaccess=abs

Danes, S.M., & Hira, T.K. (1987). Money Management Knowledge of College Students. Journal of Student Financial Aid, 17(1), 4–16.

May I suggest a brief explanation of whether your research is relevant only in the UK, or whether it presents a similar problem in another country or continent? The focus of their research may thus change, since it is possible that behaviour related to money management is not only a problem related to higher education.

I also recommend thinking about whether the revealed "problem" as financial pressure really existed from the 1980s to 2023, which is the time when the sampling studies were made.

Based on their methodology, I would like to explain why only 12 of the 789 identified studies met the selection criteria. It is recommended to think about the possibility of conducting the research in other languages (not only English) and involving the appropriate studies in their research. The revealed correlates and categories can form the basis of empirical research in the future.

I also recommend a more detailed discussion of the problem that the parental model and the opinion of peers have a positive effect on the financial behaviour of university students, because according to European research experience, the behaviour and opinion of parents and peers have a negative effect on the appropriate financial behaviour. This is confirmed by e.g. also lines 285-287.

After these changes, I recommend the article for acceptance.

6. PLOS authors have the option to publish the peer review history of their article (what does this mean?). If published, this will include your full peer review and any attached files.

Reviewer #1: **Yes: **Botond Géza Kálmán

Reviewer #2: **Yes: **Szilárd Malatyinszki

---

## [Author Response · Author response to Decision Letter 0]

28 Jun 2024

Response to Reviewers

Reviewer #1: 

COMMENT 1. I propose a minor revision in the literature review. The authors did not refer to sources that were pioneers in the research of the topic, e.g.

Bakken, M.R. (1966). Money Management Understanding of Tenth Grade Students. University of Alberta. https://archive.org/details/Bakken1966/page/n13/mode/2up

Chen, H., & Volpe, R.P. (1998). An Analysis of Personal Financial Literacy Among College Students. Financial Services Review, 7(2), 107–128. https://doi.org/10.1016/s1057-0810(99)80006-7

Chen, H., & Volpe, R.P. (2002). Gender differences in personal financial literacy among college students. Financial Services Review, 11(3), 289–308. https://go.gale.com/ps/i.do?p=AONE&sw=w&issn=10570810&v=2.1&it=r&id=GALE%7CA149166047&sid=googleScholar&linkaccess=abs

Danes, S.M., & Hira, T.K. (1987). Money Management Knowledge of College Students. Journal of Student Financial Aid, 17(1), 4–16.

Response to Comment 1: We were a little unsure whether the reviewer was referring to the scoping review itself (i.e., the Results section) or the Introduction here.

With regards to the scoping review itself, we focused our review on university students’ money management behaviours. Thus, Bakken’s (1966) study of tenth grade students was not included in our review. The studies by Chen & Volpe (1998), Chen & Volpe (2002) and Danes & Hirs (1987) did not meet our criteria for inclusion either, as they focus on financial knowledge but do not provide any measures of money management behaviours.

We recognise that the latter three are important studies of university students however. and now cite them in the Introduction (refs 24-26). See p5:

“much of the evidence regarding students’ money-management has focused on financial knowledge or literacy as outcomes of interest, rather than on specific financial behaviours (24,25,26).”

COMMENT 2: I also recommend a more detailed discussion of the problem that the parental model and the opinion of peers have a positive effect on the financial behavior of university students, because according to European research experience, the behavior and opinion of parents and peers have a negative effect on the appropriate financial behavior. This is confirmed by e.g. also lines 285-287.

After these changes, I recommend the article for acceptance.

Response to Comment 2: We have expanded coverage of social influences in the Discussion section (p29). We highlight inconsistencies between findings from quantitative and qualitative studies regarding whether parents and peers play positive or negative roles in shaping money management behaviour, and call for further research to examine what shapes the direction of such relationships:

“Syntheses of quantitative data showed that variables relating to financial beliefs and knowledge, attitudes, self-efficacy and control, and parental and peer social influence were predominantly related with ‘healthier’ money management practices, so should be harnessed as mechanisms for interventions to support students to better manage their money. Relationships between these variables and money-management practices appeared inconsistent, however. This may in part be because quantitative methodologies can oversimplify potentially complex phenomena. For example, parental norms and communication with parents and peers were each found to be positively associated with ‘better’ money management among students in quantitative studies, yet qualitative studies provided examples of students deliberately adopting positive money-management practices in response to their perceptions that their parents had managed their money badly (43). Similarly, peer influence can have an adverse influence on money management when students engage in spending behaviours aimed at matching their peers, to maintain social standing (39). The discrepancy in findings across studies underscores the complexity of social influences on money management behaviour (39, 54), and suggests that further research is needed to explore what determines whether others will have a positive or negative impact on money management.”

Reviewer #2: 

COMMENT 1. I propose a minor revision in the literature review. The authors did not refer to sources that were pioneers in the research of the topic, e.g.

Bakken, M.R. (1966). Money Management Understanding of Tenth Grade Students. University of Alberta. https://archive.org/details/Bakken1966/page/n13/mode/2up

Chen, H., & Volpe, R.P. (1998). An Analysis of Personal Financial Literacy Among College Students. Financial Services Review, 7(2), 107–128. https://doi.org/10.1016/s1057-0810(99)80006-7

Chen, H., & Volpe, R.P. (2002). Gender differences in personal financial literacy among college students. Financial Services Review, 11(3), 289–308. https://go.gale.com/ps/i.do? p=AONE&sw=w&issn=10570810&v=2.1&it=r&id=GALE%7CA149166047&sid=googleScholar&linkaccess=abs

Danes, S.M., & Hira, T.K. (1987). Money Management Knowledge of College Students. Journal of Student Financial Aid, 17(1), 4–16.

Response to Comment 1: See our response to Reviewer 1 Comment 1.

COMMENT 2. May I suggest a brief explanation of whether your research is relevant only in the UK, or whether it presents a similar problem in another country or continent? The focus of their research may thus change, since it is possible that behaviour related to money management is not only a problem related to higher education.

Response to Comment 2: We acknowledge that University students worldwide are facing financial pressure. In our original submission, we referred only to the UK and US. We now acknowledge this more explicitly by referring to research from Australia and Asia (Bangladesh) in our introduction (p4):

“University students worldwide are facing increasing financial pressure. In the UK, higher education has become substantially more expensive in recent decades, with students leaving university with significantly higher debt than at the turn of the century (1,2,3). Notable changes over this period have included increasing tuition fees for UK students (4,5), higher inflation and increased cost of living (6,7). Growing concerns around student living costs, tuition fees and loan burdens have also been documented in Australia and Asia (8,9). In the US, it has been argued that rising debt levels among graduates, who typically take student loans, may both discourage potential applicants and motivate graduates to seek career paths based on speedier debt repayment (10).”

Regarding a potential change of focus of our research, we recognise that money management is important in contexts other than higher education, and that the problems associated with money management among students in higher education may be symptomatic of broader factors. However, these issues are outside of the scope of this review, which is designed to focus solely on psychological factors surrounding money management in university students.

COMMENT 3. I also recommend thinking about whether the revealed "problem" as financial pressure really existed from the 1980s to 2023, which is the time when the sampling studies were made.

Response to Comment 3: Financial pressure among students has always been a problem. However, there have been many significant changes in the last few decades to tuition fees, student loan structures, grants, financial tools, the rise of living costs, and so on, which all may have had a significant impact on how students manage their money and the financial stress they face. Given these changes have occurred gradually over time, it is not easy to identify a specific cutoff date to reflect these changes. However, for our review findings to be relevant to the present day, date limits had to be set. We selected 2000 as the earliest publication date because, prior to 2000, in the UK at least students were entitled to grants, whereas from 2000 onwards, university students had to self-fund or take student loans. Such changes likely significantly impacted how students manage their finances, making earlier studies less reflective of the current realities and challenges faced by today's university students.

COMMENT 4. Based on their methodology, I would like to explain why only 12 of the 789 identified studies met the selection criteria.

Response to Comment 4. We have updated the PRISMA chart (Figure 1) to include explanations why the 789 studies identified by our search were reduced to 12.

COMMENT 5. It is recommended to think about the possibility of conducting the research in other languages (not only English) and involving the appropriate studies in their research. The revealed correlates and categories can form the basis of empirical research in the future.

Response to Comment 5. We conducted the review in English and German because these are the only languages spoken among the review team. This is in keeping with Cochrane Handbook (Higgins et al., 2023), which, while acknowledging the benefits of include studies irrespective of language, recognises that the review process must be pragmatic and will be limited by the linguistic expertise within the research team. 

We note that other scoping reviews published by PLoS ONE have been based on searches conducted only in languages spoken by the research team. For example, Ayala et al’s (2021) review of peer- and community-led responses to HIV was based on a search of English-only papers ‘due to resource limitations’ (; https://doi.org/10.1371/journal.pone.0260555). Similarly, Weber’s (2024) scoping review of measures of trust in physicians was based on an English-only search (https://doi.org/10.1371/journal.pone.0303840). 

COMMENT 6. I also recommend a more detailed discussion of the problem that the parental model and the opinion of peers have a positive effect on the financial behaviour of university students because, according to European research experience, the behaviour and opinion of parents and peers have a negative effect on the appropriate financial behaviour. This is confirmed by e.g. also lines 285-287.

After these changes, I recommend the article for acceptance.

Response to Comment 6: See our Response to Reviewer 1 Comment 2.

---

## [Decision Letter · Decision Letter 1]

2 Jul 2024

Understanding money-management behaviour and its potential determinants among undergraduate students: A scoping review

PONE-D-24-18530R1

Dear Dr. Gardner,

We’re pleased to inform you that your manuscript has been judged scientifically suitable for publication and will be formally accepted for publication once it meets all outstanding technical requirements.

Kind regards,

Botond Géza Kálmán, PhD

Academic Editor

PLOS ONE

Additional Editor Comments (optional):

Reviewers' comments:

Reviewer's Responses to Questions

**Comments to the Author**

1. If the authors have adequately addressed your comments raised in a previous round of review and you feel that this manuscript is now acceptable for publication, you may indicate that here to bypass the “Comments to the Author” section, enter your conflict of interest statement in the “Confidential to Editor” section, and submit your "Accept" recommendation.

Reviewer #1: (No Response)

Reviewer #2: All comments have been addressed

2. Is the manuscript technically sound, and do the data support the conclusions?

Reviewer #1: Yes

Reviewer #2: Yes

3. Has the statistical analysis been performed appropriately and rigorously? 

Reviewer #1: Yes

Reviewer #2: Yes

4. Have the authors made all data underlying the findings in their manuscript fully available?

Reviewer #1: Yes

Reviewer #2: Yes

5. Is the manuscript presented in an intelligible fashion and written in standard English?

Reviewer #1: Yes

Reviewer #2: Yes

6. Review Comments to the Author

Reviewer #1: I thank the authors for the detailed answers to the questions in the comments and for the scientific revision of the article. Taking into account the changes and responses, I recommend the publication of the article.

Reviewer #2: (No Response)

7. PLOS authors have the option to publish the peer review history of their article (what does this mean?). If published, this will include your full peer review and any attached files.

Reviewer #1: **Yes: **Botond Géza KÁLMÁN

Reviewer #2: **Yes: **Szilárd Malatyinszki

---

## [Editor Report · Acceptance letter]

3 Jul 2024

PONE-D-24-18530R1 

PLOS ONE

Dear Dr. Gardner, 

I'm pleased to inform you that your manuscript has been deemed suitable for publication in PLOS ONE. Congratulations! Your manuscript is now being handed over to our production team.

Kind regards, 

on behalf of

Dr. Botond Géza Kálmán 

Academic Editor

PLOS ONE